# PandaLM: An Automatic Evaluation Benchmark for LLM Instruction Tuning Optimization

**Yidong Wang**[1,2*], **Zhuohao Yu**[1*],
**Wenjin Yao**[1], **Zhengran Zeng**[1], **Linyi Yang**[2], **Cunxiang Wang**[2],
**Hao Chen**[3], **Chaoya Jiang**[1] , **Rui Xie**[1], **Jindong Wang**[3], **Xing Xie**[3],
**Wei Ye**[1†], **Shikun Zhang**[1†], **Yue Zhang**[2†]

[1]Peking University   [2]Westlake University   [3]Microsoft Research Asia

## Abstract

Instruction tuning large language models (LLMs) remains a challenging task, owing to the complexity of hyperparameter selection and the difficulty involved in evaluating the tuned models. To determine the optimal hyperparameters, an automatic, robust, and reliable evaluation benchmark is essential. However, establishing such a benchmark is not a trivial task due to the challenges associated with evaluation accuracy and privacy protection. In response to these challenges, we introduce a judge large language model, named PandaLM, which is trained to distinguish the superior model given several LLMs. PandaLM's focus extends beyond just the objective correctness of responses, which is the main focus of traditional evaluation datasets. It addresses vital subjective factors such as relative conciseness, clarity, adherence to instructions, comprehensiveness, and formality. To ensure the reliability of PandaLM, we collect a diverse human-annotated test dataset, where all contexts are generated by humans and labels are aligned with human preferences. On evaluations using our collected test dataset, our findings reveal that PandaLM-7B offers performance comparable to both GPT-3.5 and GPT-4. Impressively, PandaLM-70B surpasses their performance. PandaLM enables the evaluation of LLM to be fairer but with less cost, evidenced by significant improvements achieved by models tuned through PandaLM compared to their counterparts trained with default Alpaca's hyperparameters. In addition, PandaLM does not depend on API-based evaluations, thus avoiding potential data leakage.

## 1 Introduction

Large language models (LLMs) have attracted increasing attention in the field of artificial intelligence (OpenAI, 2023; Google, 2023; Zeng et al., 2022a; Brown et al., 2020; Chowdhery et al., 2022; Anil et al., 2023; Zhang et al., 2023), with various applications from question answering (Hirschman & Gaizauskas, 2001; Kwiatkowski et al., 2019; Wang et al., 2021), machine translation (Vaswani et al., 2017; Stahlberg, 2020) to content creation (Biswas, 2023; Adams & Chuah, 2022). The Alpaca project (Taori et al., 2023) has been a pioneering effort in instruction tuning of LLaMA (Touvron et al., 2023), setting a precedent for instruction tuning LLMs, followed by Vicunna (Chiang et al., 2023). Subsequent research (Diao et al., 2023; Ji et al., 2023; Chaudhary, 2023) have typically adopted Alpaca's hyperparameters as a standard for training their LLMs. Given the necessity of instruction tuning for these pre-trained models to effectively understand and follow natural language instructions (Wang et al., 2022c; Taori et al., 2023; Peng et al., 2023), optimizing their tuning hyperparameters is crucial for peak performance. Critical factors such as optimizer selection, learning rate, number of training epochs, and quality and size of training data significantly influence the model's performance (Liaw et al., 2018; Tan & Le, 2019). However, a research gap remains in the area of hyperparameter optimization specifically designed for instruction tuning LLMs. To address this issue,

---

*Equal contribution. Yidong did this work during his internship at Westlake University.
†Corresponding to wye@pku.edu.cn; zhangsk@pku.edu.cn; zhangyue@westlake.edu.cn.

we aim to construct an automated, reliable, and robust evaluation method, which can be integrated into any open-sourced LLMs and used as the judging basis for hyperparameter optimization.

The development of such an evaluation method presents its challenges (Guo et al., 2023; Chang et al., 2023), including ensuring evaluation reliability and privacy protection. In the context of our paper, when we refer to "privacy", we primarily allude to the principles ingrained in federated learning (Zhang et al., 2021a) which enables model training across multiple devices or servers while keeping the data localized, thus offering a degree of privacy. Current methods often involve either crowd-sourcing work or API usage, which could be costly, and time-consuming. Besides, these methods face challenges in terms of consistency and reproducibility. This is primarily due to the lack of transparency regarding language model change logs and the inherent subjectivity of human annotations. Note that utilizing API-based evaluations carries the risk of potentially high costs associated with addressing data leaks. Although open-sourced LLMs can be alternative evaluators, they are not specifically designed for assessment, thus making it difficult to deploy them directly as evaluators.

On the other hand, the labels of previous evaluation methods (Zheng et al., 2023; Gao et al., 2021; Wang et al., 2023b;c) simply definite answers and fail to consider the language complexity in practice. The evaluation metrics of these procedures are typically accuracy and F1-score, without considering the subjective evaluation metrics that autoregressive generative language models should pay attention to, thus not reflecting the potential of such models to generate contextually relevant text. The appropriate subjective evaluation metrics can be relative conciseness, clarity, adherence to instructions, comprehensiveness, formality, and context relevance.

To tackle these challenges, we introduce a judge language model, aiming for **Re**producible **and A**utomated **L**anguage **M**odel Assessment (PandaLM). Tuned from LLaMA, PandaLM is used to distinguish the most superior model among various candidates, each fine-tuned with different hyperparameters, and is also capable of providing the rationale behind its choice based on the reference response for the context. PandaLM surpasses the limitations of traditional evaluation methods and focuses on more subjective aspects, such as relative conciseness, clarity, comprehensiveness, formality, and adherence to instructions. Furthermore, the robustness of PandaLM is strengthened by its ability to identify and rectify problems such as logical fallacies, unnecessary repetitions, grammatical inaccuracies, and context irrelevance. By considering these diverse aspects, we leverage PandaLM's ability to distinguish the most superior model among candidates on the validation set and then provide insights for facilitating hyperparameter optimization of instruction tuning.

In practice, we generate paired responses from a diverse set of similarly sized foundation models including LLaMA-7B (Touvron et al., 2023), Bloom-7B (Scao et al., 2022), Cerebras-GPT-6.7B (Dey et al., 2023), OPT-7B (Zhang et al., 2022a), and Pythia-6.9B (Biderman et al., 2023). Each of these models is fine-tuned using the same data and hyperparameters as Alpaca (Taori et al., 2023). The paired responses from these tuned LLMs constitute the input of training data for PandaLM. The most straightforward approach to generate the corresponding target of training data is through human annotation, but this method can be costly and time-consuming (Wang et al., 2023h). And the lack of sufficiently annotated training data has always been a significant issue in the era of deep learning Ouali et al. (2020); Wang et al. (2023e); Zhang et al. (2021b); Chen et al. (2023); Wang et al. (2022a). Considering that GPT-3.5 can provide a reliable evaluation to some extent, to reduce costs, we follow self-instruct (Wang et al., 2022c), which is a methodology that capitalizes on pre-existing knowledge within large language models to generate annotations or outputs through self-generated instructions. to distil data from GPT-3.5 and apply heuristic data filtering strategies to mitigate noise. Specifically, we filter out invalid evaluations from gpt-3.5-turbo with hand-crafted rules. To address position bias, we also filter out inconsistent samples when swapping the orders of responses in the prompt. Despite the utilization of data distilled from GPT-3.5, the active removal of noise enhances the quality of the training data, fostering a more efficient and robust training process for PandaLM.

To ensure the reliability of PandaLM, we develop a test dataset that aligns with human preference and covers a wide range of tasks and contexts. The instructions and inputs of test data are sampled from the human evaluation dataset of self-instruct (Wang et al., 2022c), with responses generated by different LLMs and each label independently provided by three different human evaluators. Samples with significant divergences are excluded to ensure the Inter Annotator Agreement (IAA) of each annotator remains larger than 0.85. As illustrated in Table 2, PandaLM-7B showcases robust and competitive performance. Remarkably, the efficacy of PandaLM-70B is even more pronounced,

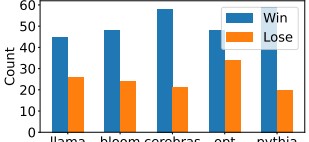 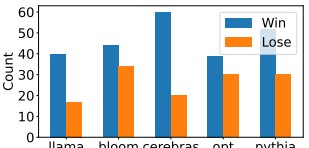 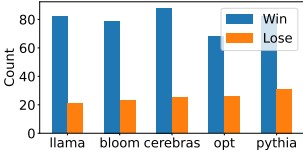

(a) Comparison Results of GPT-3.5.  (b) Comparison Results of GPT-4.  (c) Comparison Results of Human.

Figure 1: The models are evaluated and compared using both GPT-3.5, GPT-4 and human annotators. The 'Win' count represents the number of responses where models fine-tuned with PandaLM-selected optimal hyperparameters outperform models using Alpaca's hyperparameters. Conversely, the 'Lose' count represents the number of responses where models utilizing Alpaca's hyperparameters produce superior responses compared with those fine-tuned with the optimal hyperparameters determined by PandaLM. Note that the overall test set comprises 170 instances, and 'Tie' scenarios are not considered in this illustration.

exceeding the performance metrics of GPT-4. This enhancement is largely attributable to the effective noise mitigation strategies employed during the training phase.

Moreover, as illustrated in Figure 1, adopting PandaLM's selected optimal hyperparameters covering optimizer selection, learning rate, number of training epochs, and learning rate scheduler brings noteworthy improvements. When assessed using GPT-4 with a set of 170 instructions, a group of five open language models, tuned with optimal hyperparameters selected by PandaLM, achieves an average of 47.0 superior responses and 26.2 inferior responses, outperforming those trained using Alpaca's hyperparameters. *Note that the training data remains the same for conducting fair comparisons.* Moreover, when these LLMs are evaluated by human experts, using the same set of 170 instructions, they exhibit an average of 79.8 superior responses and 25.2 inferior responses, once again surpassing the performance of models trained with Alpaca's hyperparameters. The experimental results underline the effectiveness of PandaLM in determining optimal hyperparameters for choosing the best LLMs. In addition, when the fine-tuned LLMs are assessed using the lm-eval (Gao et al., 2021), a unified framework to test LLM on a large number of different traditional evaluation tasks, the results further reinforce the superiority of LLMs optimized by PandaLM.

In conclusion, our work delivers three key contributions:

- We introduce PandaLM, a privacy-protected judge language model for evaluating and optimizing hyperparameters for LLMs.

- We create a reliable human-annotated dataset, essential for validating PandaLM's performance and further research.

- We make use of PandaLM to optimize the hyperparameters of a series of open-sourced LLMs. In comparison to those LLMs tuned using hyperparameters identified by Alpaca, tuning models with PandaLM-selected hyperparameters yields substantial performance enhancements.

## 2 RELATED WORK

This section reviews the relevant literature on the topic of hyperparameter optimization and the evaluation of language models.

**Hyperparameter Optimization** The importance of hyperparameter optimization in machine learning (Yu & Zhu, 2020; Falkner et al., 2018; Li et al., 2017; Xu et al., 2023; Wang et al., 2023a; Wu et al., 2019), particularly in the context of fine-tuning deep learning language models such as BERT (Kenton & Toutanova, 2019) and GPT (Radford et al.), cannot be ignored. For these models, the choice of hyperparameters like the learning rate, batch size, or the number of training epochs can significantly influence their performance (Godbole et al., 2023; Sun et al., 2019; Tunstall et al., 2022). This selection process becomes even more critical when fine-tuning these models on domain-specific tasks,

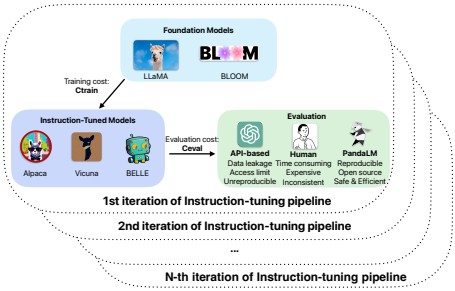

Figure 2: The pipeline of instruction tuning LLMs.

where the optimal set of hyperparameters can vary significantly among different domains (Dodge et al., 2020; Sun et al., 2019).

**Evaluation of Language Models** Accurate evaluation of language models is crucial in determining optimal hyperparameters, thus improving the models' overall performance (Sun et al., 2019; Godbole et al., 2023). Conventional objective metrics like perplexity (Mallio et al., 2023) and accuracy (Xu et al., 2020; Wang et al.; Yang et al., 2022a; Zhong et al., 2023) on downstream tasks (Gao et al., 2021) provide valuable insights, but they may not effectively guide the choice of hyperparameters to enhance LLMs (Rogers et al., 2021) because evaluating LLMs requires other subjective metrics. Advanced language models, such as GPT-4 (OpenAI, 2023) and Bard (Google, 2023), incorporate human evaluations as part of their testing method for LLMs, aiming to better align with human judgements (Wang et al., 2023h). Although human-based evaluation methods offer considerable insight into a model's performance, they are costly and labor-intensive, making it less feasible for iterative hyperparameter optimization processes. Recent advancements in NLP have brought forth model-based metrics such as BERTScore (Zhang et al., 2019) and MAUVE (Pillutla et al., 2021). While these metrics offer valuable insights, there are significant areas where they may not align perfectly with the objectives of response evaluation. Firstly, BERTScore and MAUVE are engineered to measure the similarity between generated content and reference text. However, they are not inherently designed to discern which of multiple responses is superior. A response is closer to a human-written reference doesn't necessarily mean it adheres better to given instructions or satisfies a specific context. Secondly, while these metrics yield scores that represent content similarity, they aren't always intuitive to users. Interpreting these scores and translating them into actionable feedback can be a challenge. In contrast, PandaLM offers a more straightforward approach. It is tailored to directly output the evaluation result in an interpretable manner, making the feedback process transparent and easily understood by humans. In conclusion, while metrics like BERTScore and MAUVE provide valuable insights into content similarity, there is a pressing need for specialized evaluation tools like PandaLM. Tools that not only discern response quality but also do so in a user-friendly, human-comprehensible manner.

Subjective qualitative analysis of a model's outputs, such as its ability to handle ambiguous instructions and provide contextually appropriate responses, is increasingly being recognized as a valuable metric for evaluating models (Zheng et al., 2023). Optimizing hyperparameters with considerations towards these qualitative measures could lead to models that perform more robustly in diverse real-world scenarios. The previous qualitative analysis can be achieved either through human evaluators or through APIs of advanced language models, which is different from our motivation.

## 3 METHODOLOGY

As shown in Figure 2, the process of instruction tuning begins with a foundation model, which is then fine-tuned using instructions. The performance of each tuned model is evaluated to determine the best output. This involves exploring numerous models, each tuned with different hyperparameters, to identify the optimal one. To facilitate this pipeline, a reliable and automated language model assessment system is essential. To address this, we introduce PandaLM - a judge LLM specifically designed to assess the performance of LLMs fine-tuned with various parameters. Our goal is to identify the superior model from a pool of candidates accurately.

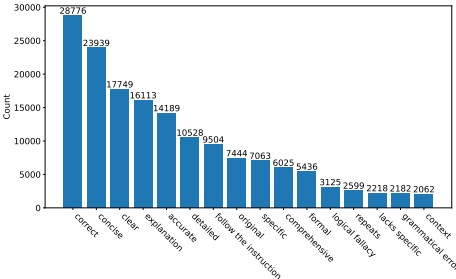

Figure 3: The top 16 words used in the PandaLM-7B evaluation reasons from randomly sampled 80k evaluation outputs. An example of evaluation reason and evaluation outputs can be found in Figure 5. Stop words are filtered.

## 3.1 TRAIN DATA COLLECTION AND PREPROCESSING

The training data collection aims to create a rich dataset that allows the model to evaluate different responses in a given context and generate an evaluation reason and a reference response using the same context. As demonstrated in Appendix A, each training data instance consists of an input tuple (instruction, input, response1, response2) and an output tuple (evaluation result, evaluation reason, reference response). The instructions and inputs in the input tuple are sampled from the Alpaca 52K dataset (Taori et al., 2023). The response pairs are produced by various instruction-tuned models: LLaMA-7B (Touvron et al., 2023), Bloom-7B (Scao et al., 2022), Cerebras-GPT-6.7B (Dey et al., 2023), OPT-7B (Zhang et al., 2022a), and Pythia-6.9B (Biderman et al., 2023). These models are selected due to their comparable sizes and the public availability of their model weights. Each is fine-tuned using the same instruction data and hyperparameters following Alpaca (Taori et al., 2023). The corresponding output tuple includes an evaluation result, a brief explanation for the evaluation, and a reference response. The evaluation result would be either '1' or '2', indicating that response 1 or response 2 is better, and 'Tie' indicates that two responses are similar in quality. The training prompt of PandaLM is shown at Appendix A.As it is impractical to source millions of output tuples from human annotators, and given that GPT-3.5 is capable of evaluating LLMs to some degree, we follow self-instruct (Wang et al., 2022c) to generate output tuples using GPT-3.5. As illustrated in Figure 3, we design prompts carefully to guide the generation of training data for PandaLM. The goal is to ensure PandaLM not only prioritizes objective response correctness but also emphasizes critical subjective aspects such as relative conciseness, clarity, comprehensiveness, formality, and adherence to instructions. Besides, we encourage PandaLM to identify and rectify issues like logical fallacies, unnecessary repetitions, grammatical inaccuracies, and the absence of context relevance. A heuristic data filtering strategy is then applied to remove noisy data. Specifically, to address the observed inherent bias in GPT-3.5 regarding the order of input responses even with carefully designed prompts, samples from the training dataset are removed if their evaluation results conflict when the orders of input responses are swapped. We finally obtain a filtered dataset containing 300K samples.

## 3.2 PANDALM TRAINING

In this subsection, we provide details about the training procedure for PandaLM. The backbone of PandaLM is LLaMA model, as it exhibits strong performance on multiple complicated NLP tasks (Beeching et al., 2023).

During the fine-tuning phase of PandaLM, we use the standard cross-entropy loss targeting the next token prediction. The model operates in a sequence-to-sequence paradigm without the necessity for a separate classification head. We train PandaLM with the DeepSpeed (Rasley et al., 2020) library, and Zero Redundancy Optimizer (ZeRO) (Rajbhandari et al., 2020; Ren et al., 2021) Stage 2, on 8 NVIDIA A100-SXM4-80GB GPUs. We use the bfloat16 (BF16) computation precision option to further optimize the model's speed and efficiency. Regarding the training hyperparameters, we apply the AdamW (Loshchilov & Hutter, 2017) optimizer with a learning rate of 2e-5 and a cosine learning rate scheduler. The model is trained for 2 epochs. The training process uses a warmup ratio of 0.03 to

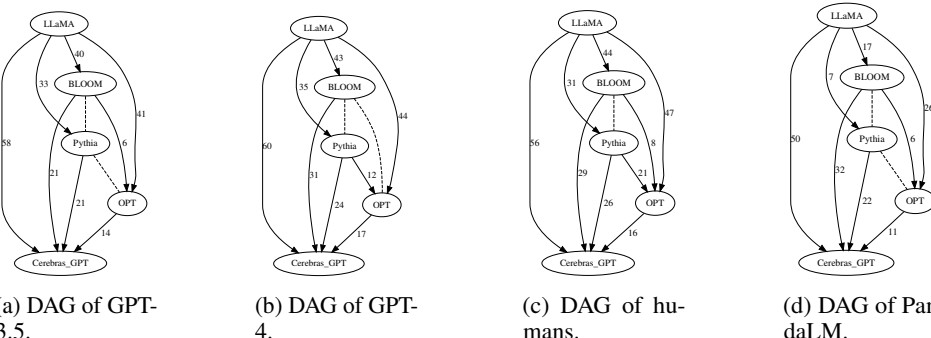

| (a) DAG of GPT-3.5. | (b) DAG of GPT-4. | (c) DAG of humans. | (d) DAG of PandaLM. |

Figure 4: Comparative Visualization of Model Performance. The instruction-tuned models use the same training data and hyperparameters. A directed edge from node A to B indicates model A's significant superiority over B, while a dashed undirected edge indicates the two models are similar in performance. The number associated with the directed edge (A, B) represents the difference between the number of wins and losses for model A compared to model B. The absence of a number on the dashed undirected edge indicates that the difference between the number of wins and losses for the models is smaller than 5. We swap the order of two responses to perform inference twice on each data. The conflicting evaluation results are then modified to 'Tie'.

avoid large gradients at the beginning of training. We use a batch size of 2 per GPU with all inputs truncated to a maximum of 1024 tokens and employ a gradient accumulation strategy with 8 steps.

## 4 RELIABILITY EVALUATION OF PANDALM

To ensure the reliability of PandaLM, we create a test dataset that is labeled by humans and designed to align with human preferences for responses. Each instance of this test dataset consists of one instruction and input, and two responses produced by different instruction-tuned LLMs. The paired responses are provided by LLaMA-7B, Bloom-7B, Cerebras-GPT-6.7B, OPT-7B, and Pythia-6.9B, all instruction tuned using the same instruction data and hyperparameters following Alpaca (Taori et al., 2023). The test data is sampled from the diverse human evaluation dataset of self-instruct (Wang et al., 2022c), which includes data from Grammarly, Wikipedia, National Geographic and nearly one hundred apps or websites. The inputs and labels are solely human-generated and include a range of tasks and contents. Three different human evaluators independently annotate the labels indicating the preferred response. Samples with significant divergences are excluded to ensure the Inter Annotator Agreement (IAA) of each annotator remains larger than 0.85. This is because such samples demand additional knowledge or hard-to-obtain information, making them challenging for humans to evaluate. The filtered test dataset contains 1K samples, while the original unfiltered dataset has 2.5K samples.

To maintain high-quality crowdsourcing work, we involve three experts to annotate the same data point concurrently during the annotation process. There is no prior relationship between the experts and the authors. The experts are hired from an annotation company. These experts receive specialized training that goes beyond evaluating response correctness, enabling them to emphasize other crucial aspects like relative conciseness, clarity, comprehensiveness, formality, and adherence to instructions. Furthermore, we guide these annotators in identifying and addressing issues such as logical fallacies, unnecessary repetitions, grammatical inaccuracies, and a lack of contextual relevance. All human ratings are collected consistently within the same session. To ensure clarity and consistency, we provide comprehensive instructions for every annotator. After the trial phase of data annotation, we eliminate some low-quality labeled data. The final IAA amongst the three annotators, as measured by Cohen's Kappa (Cohen, 1960), yields average scores of 0.85, 0.86, and 0.88 respectively, indicating a relatively high level of reliability for our test dataset. To refine the model's performance assessment compared to human evaluators, we can use the inter-annotator agreement (IAA) of 0.85 as a benchmark. If our model exceeds this, it indicates strong performance. However, setting a realistic target slightly above this human IAA, say around 0.90, offers a challenging yet achievable goal. The distribution of the test data comprises 105 instances of ties, 422 instances where Response 1 wins, and 472 instances where Response 2 takes the lead. Note that the human-generated dataset has no

Table 1: Comparative analysis of evaluation results from various annotation models. The tuple in the table means (#win,#lose,#tie). Specifically, (72,28,11) in the first line of the table indicates that LLaMA-7B outperforms Bloom-7B in 72 responses, underperforms in 28, and matches the quality in 11 responses. The 'Judged By' column represents different methods of response evaluation. 'Human' indicates that humans evaluate the result, and 'PandaLM' indicates that our proposed PandaLM model evaluates the result.

| Judged By | Base Model | LLaMA-7B | Bloom-7B | Cerebras-6.7B | OPT-7B | Pythia-6.9B |
|---|---|---|---|---|---|---|
| Human | LLaMA-7B | / | (72,28,11) | (80,24,6) | (71,24,11) | (58,27,9) |
| | Bloom-7B | (28,72,11) | / | (59,30,11) | (43,35,11) | (47,49,11) |
| | Cerebras-6.7B | (24,80,6) | (30,59,11) | / | (33,49,9) | (27,53,11) |
| | OPT-7B | (24,71,11) | (35,43,11) | (49,33,9) | / | (32,53,15) |
| | Pythia-6.9B | (27,58,9) | (49,47,11) | (53,27,11) | (53,32,15) | / |
| GPT-3.5 | LLaMA-7B | / | (59,19,33) | (71,13,26) | (58,17,31) | (49,16,29) |
| | Bloom-7B | (19,59,33) | / | (40,19,41) | (36,30,23) | (33,34,40) |
| | Cerebras-6.7B | (13,71,26) | (19,40,41) | / | (24,38,29) | (22,43,26) |
| | OPT-7B | (17,58,31) | (30,36,23) | (38,24,29) | / | (30,30,40) |
| | Pythia-6.9B | (16,49,29) | (34,33,40) | (43,22,26) | (30,30,40) | / |
| GPT-4 | LLaMA-7B | / | (58,15,38) | (69,9,32) | (58,14,34) | (52,17,25) |
| | Bloom-7B | (15,58,38) | / | (47,16,37) | (35,31,23) | (32,33,42) |
| | Cerebras-6.7B | (9,69,32) | (16,47,37) | / | (23,40,28) | (17,41,33) |
| | OPT-7B | (14,58,34) | (31,35,23) | (40,23,28) | / | (25,37,38) |
| | Pythia-6.9B | (17,52,25) | (33,32,42) | (41,17,33) | (37,25,38) | / |
| PandaLM-7B | LLaMA-7B | / | (46,29,36) | (68,18,24) | (52,26,28) | (35,28,31) |
| | Bloom-7B | (29,46,36) | / | (50,18,32) | (36,30,23) | (36,31,40) |
| | Cerebras-6.7B | (18,68,24) | (18,50,32) | / | (28,39,24) | (24,46,21) |
| | OPT-7B | (26,52,28) | (30,36,23) | (39,28,24) | / | (30,32,38) |
| | Pythia-6.9B | (28,35,31) | (31,36,40) | (46,24,21) | (32,30,38) | / |

Table 2: Comparison between Human Annotation results and Judged Model evaluation results.

| Judged Model | Accuracy | Precision | Recall | F1 |
|---|---|---|---|---|
| GPT-3.5 | 0.6296 | 0.6195 | 0.6359 | 0.5820 |
| GPT-4 | 0.6647 | 0.6620 | **0.6815** | 0.6180 |
| PandaLM-7B | 0.5926 | 0.5728 | 0.5923 | 0.5456 |
| PandaLM-70B-LoRA | 0.6186 | **0.7757** | 0.6186 | 0.6654 |
| PandaLM-70B | **0.6687** | 0.7402 | 0.6687 | **0.6923** |

personally identifiable information or offensive content, and all annotators receive redundant labor fees.

After obtaining the human-labeled test dataset, we can assess and compare the evaluation performances of GPT-3.5, GPT-4, and PandaLM. An interesting observation from Table 1 is the shared similar partial order graph between GPT-3.5, GPT-4, PandaLM-7B, and humans. Furthermore, Figure 4 illustrates directed orders of model superiority (if model A outperforms model B, a directed edge from A to B is drawn; if model A and model B perform similarly, a dashed line from A to B is drawn.), and provides a visual representation of comparative model effectiveness. The experimental results indicate similarities in the preferences of GPT-3.5, GPT-4, PandaLM-7B, and humans. *Note that for PandaLM, GPT-3.5, and GPT-4, we swap the input response order and infer twice to procure the final evaluation output. The conflicting evaluation results are revised to 'Tie'.*

As shown in Table 2, we conduct a statistical analysis comparing the accuracy, precision, recall, and F1-score of GPT-3.5, GPT-4, and PandaLM against human annotations. The performance of PandaLM-70B even surpasses that of GPT-4. The results indicate the efficacy of removing noise in training data and the choosing of foundational model architectures and instructions.

To prove the robustness and adaptability of PandaLM across distribution shifts, we also concentrate our evaluations on distinct areas, with a particular emphasis on the legal (LSAT) and biological (PubMedQA and BioASQ) domains. The introduction of the used datasets can be found at Appendix D. Note that for generating responses, we employ open-sourced language models such as Vicuna and Alpaca. Due to constraints in time, GPT-4 is adopted to produce the gold standard answers (win/tie/lose of responses) instead of human annotators. The results illustrated in Table 3 underscore PandaLM's prowess not only in general contexts but also in specific domains such as law (via LSAT) and biology (via PubMedQA and BioASQ). Since we are addressing a three-category classification task (win/lose/tie), where a random guess would lead to around 33% in the precision, recall, and F1 score. PandaLM-7B's results are notably above this level. To further validate PandaLM-7B, we conducted a human evaluation with 30 samples on the BioASQ evaluation. The human evaluation showed that both PandaLM-7B and GPT-4 tended to favor Vicuna over Alpaca, indicating a consistent trend in their evaluations. The marked improvement in performance as the model scales up reaffirms PandaLM's promise across varied applications, further emphasizing its reliability amidst different content distributions.

We further investigate the performance of PandaLM by contrasting its efficacy when trained solely on numerical comparisons (win/tie/lose) — akin to a traditional reward model — with the holistic

Table 3: Performance evaluation of PandaLM across diverse domains. The table showcases the accuracy, precision, recall, and F1 scores achieved by PandaLM of two different sizes (finetuned from LLaMA-7B and LLaMA2-70B) on three distinct datasets: LSAT, PubMedQA, and BioASQ. These datasets are representative of the legal and biological domains, chosen to demonstrate the robustness and adaptability of PandaLM to different distribution shifts. It's worth noting that GPT-4 was employed for generating the gold standard answers instead of human annotations.

| | Accuracy | Precision | Recall | F1 Score |
|---|---|---|---|---|
| LSAT (PandaLM-7B) | 0.4717 | 0.7289 | 0.4717 | 0.5345 |
| LSAT (PandaLM-70B) | 0.6604 | 0.7625 | 0.6604 | 0.6654 |
| PubMedQA (PandaLM-7B) | 0.6154 | 0.8736 | 0.6154 | 0.6972 |
| PubMedQA (PandaLM-70B) | 0.7692 | 0.7811 | 0.7692 | 0.7663 |
| BioASQ (PandaLM-7B) | 0.5152 | 0.7831 | 0.5152 | 0.5602 |
| BioASQ (PandaLM-70B) | 0.7727 | 0.8076 | 0.7727 | 0.7798 |

Table 4: Comparison of PandaLM performance w/ and w/o reasons and references.

| | Accuracy | Precision | Recall | F1 Score |
|---|---|---|---|---|
| PandaLM-7B with only win/tie/lose | 0.4725 | 0.4505 | 0.4725 | 0.3152 |
| PandaLM-7B | 0.5926 | 0.5728 | 0.5923 | 0.5456 |

Table 5: Evaluation of the effectiveness of PandaLM's selected hyperparameters and Alpaca's hyperparameters. The tuple in the table means (#win,#lose,#tie). Specifically, (45,26,99) in the first line of the table indicates that PandaLM's hyperparameter-tuned LLaMA-7B outperforms Alpaca's version in 45 responses, underperforms in 26, and matches the quality in 99 instances. The 'Judged By' column represents different methods of response evaluation.

| Judge Model | LLaMA-7B | Bloom-7B | Cerebras-6.7B | OPT-7B | Pythia-6.9B |
|---|---|---|---|---|---|
| GPT-3.5 | (45,26,99) | (48,24,98) | (58,21,91) | (48,34,88) | (59,20,91) |
| GPT-4 | (40,17,113) | (44,34,92) | (60,20,90) | (39,30,101) | (52,30,88) |
| Human | (82,21,67) | (79,23,68) | (88,25,57) | (68,26,76) | (82,31,57) |

approach of the standard PandaLM that incorporates evaluation reasons and reference responses. As shown in Table 4, it become evident that the evaluation reasons and reference responses significantly aid LLMs in understanding the evaluation tasks. Note that in Appendix G, the results clearly demonstrate that a smaller model, when precisely tuned, has the capability to outperform a larger, untuned model in evaluation metrics. This finding emphasizes the significant impact of targeted tuning on model performance in evaluation scenarios. We also provide an analysis on PandaLM across model shifts in Appendix K.

In addition, beyond performance metrics, PandaLM introduces unique advantages that are not present in models like GPT-3.5 and GPT-4. It offers open-source availability, enabling reproducibility, and protecting data privacy. Furthermore, it provides unlimited access, removing any restrictions that might hinder comprehensive evaluation and application.

## 5 USING PANDALM TO INSTRUCTION TUNE LLMS

To highlight the effectiveness of using PandaLM for instruction tuning LLMs, we compare the performance of models tuned with PandaLM's selected optimal hyperparameters against those tuned with Alpaca's parameters using GPT-3.5, GPT-4, and human experts. It is noteworthy that PandaLM-7B is employed for this comparison due to considerations regarding computational resources. Given the proven effectiveness of PandaLM-7B, there is a grounded expectation that the performance of PandaLM-70B will exhibit further enhancement. This comparison evaluates multiple tuned LLMs: LLaMA-7B, Bloom-7B, Cerebras-GPT-6.7B, OPT-7B, and Pythia-6.9B. The assessment is conducted on a validation set comprising 170 distinct instructions and inputs obtained from our 1K test set introduced in Section 4. Alpaca's tuning protocol involves training for three epochs with the final iteration's checkpoints being used. It uses the AdamW (Loshchilov & Hutter, 2017) optimizer with a learning rate of 2e-5 and a cosine learning rate scheduler. We perform a wider range of hyperparamters to tune LLMs using PandaLM-7B. Specifically, we explore checkpoints from each epoch (ranging from epoch 1 to epoch 5), four different learning rates (2e-6, 1e-5, 2e-5, 2e-4), two types of optimizers (SGD (Goodfellow et al., 2016) and AdamW), and two learning rate schedulers (cosine and linear). In total, this creates a configuration space of 80 different possibilities per model.

We search for optimal hyperparameters among the 80 configurations. These are divided into four blocks, each containing 20 configurations. Sequential comparisons identify the best configuration in each block. The top configurations from each block are then compared to determine the overall best configuration. We repeat each comparison twice for robustness and carry out 800 comparisons in total. The conflicting evaluation results are modified to 'Tie'. Key insights from our tuning process include: Bloom-7B performs best with SGD, a learning rate of 2e-5, and a cosine schedule over 5 epochs. Cerebras-GPT-6.7B also favors SGD with the same learning rate but with a linear schedule. LLaMA-7B prefers AdamW, a learning rate of 1e-5, and a linear schedule over 4 epochs. OPT-6.7B achieves top results with AdamW, a learning rate of 2e-5, and a linear scheduler over 5 epochs. Pythia-6.9B prefers SGD, a learning rate of 1e-5, a cosine schedule, and 5 epochs. This highlights the importance of customized hyperparameter tuning for different models to achieve peak performance. We also provide the analysis on data size, quality and LoRA in Appendix E and Appedix F.

As illustrated in Table 5, for GPT-3.5, GPT-4, and human, all base models achieve superior performance when tuned with PandaLM's selected hyperparameters compared to Alpaca's hyperparameters. *Note that the procedure of switching the order of input responses, as applied for PandaLM, is also implemented for GPT-3.5 and GPT-4 to acquire more robust evaluation results.* This outcome not only supports the claim that PandaLM-7B can enhance the performance of models but also highlights its potential to further improve various large language models. Besides, as shown in Appendix B, based on PandaLM's evaluation, the model demonstrating superior performance is LLaMA-PandaLM. Note that the base foundation model's characteristics can be a significant factor in performance, as evidenced by LLaMA models securing the top two positions. The ranking pattern observed aligns closely with the base model rankings presented in Figure 4. We also provide a hyperparameter optimization analysis in Appendix J.

Moreover, Table 6 in Appendix C compares fine-tuned LLMs on various traditional tasks with lm-eval (Gao et al., 2021). Interestingly, while most language models display enhanced performance with PandaLM finetuning, Cerebras experiences a dip. This underscores the value of subjective evaluation (win/tie/lose of responses), as evaluations from humans, GPT-4, and GPT-3.5 all indicate superior performance for Cerebras with PandaLM.

# 6 LIMITATIONS

While the outcomes of our study are encouraging, we discuss several limitations here. Firstly, the selected range of hyperparameters used in this work is based on common practice and prior literature, and thus may not encompass the absolute optimal hyperparameters. While extending the search bond will inevitably increase the computational cost. While the core data, derived from GPT-3.5, may not fully resonate with human preferences, it's essential to recognize that the efficacy of LLMs hinges not just on training data but also on foundational model architectures and instructions. Currently, our emphasis is primarily on outcome-based evaluation, which is indeed resource-intensive. However, integrating behavior prediction (e.g., using rational analysis Lu et al. (2022); Wang et al. (2023d; 2020); Yang et al. (2021; 2023a)) into an evaluation framework could offer a more comprehensive understanding of LLM performance. For instance, analyzing and evaluating the extended text outputs of an untuned LLM can help predict how a tuned version might behave in various scenarios. This approach could provide a more efficient and insightful way to balance resource-heavy outcome assessments. Besides, we only research on supervised training of LLMs, but the realistic data could be imbalanced or unlabeled, hence semi-supervised Wang et al. (2023e); Chen et al. (2023); Wang et al. (2022b), noisy Zhang et al. (2022b); Chen et al. (2024) and imbalanced training Yang et al. (2022b); Wang et al. (2023f;g) are also our future directions.

# 7 CONCLUSION

In our exploration of hyperparameter optimization, we apply PandaLM: an automatic and reliable judge model for the tuning of LLMs. Our findings demonstrate that the use of PandaLM is feasible and consistently produces models of superior performance compared to those tuned with Alpaca's default parameters. We are dedicated to continually enhancing PandaLM by expanding its capacity to support larger models and analyzing its intrinsic features, thereby developing increasingly robust versions of the judging model in the future.

## ACKNOWLEDGEMENT

We would like to thank the anonymous reviewers for their insightful comments and suggestions to help improve the paper. This publication has emanated from research conducted with the financial support of the Pioneer and "Leading Goose" R&D Program of Zhejiang under Grant Number 2022SDXHDX0003 and the National Natural Science Foundation of China Key Program under Grant Number 62336006.

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

```
"inputs": {
    "instruction": "Find an example of the given kind of data",
    "input": "Qualitative data",
    "response1": "An example of qualitative data is customer feedback.",
    "response2": "An example of qualitative data is a customer review."
}
"outputs": {
    "evaluation_result": "Tie",
    "evaluation_reason": "Both responses are correct and provide similar examples of qualitative data.",
    "reference_response": "An example of qualitative data is an interview transcript."
}
```

Figure 5: A training data example for PandaLM.

```
Below are two responses for a given task. The task is defined by the Instruction with an Input
that provides further context. Evaluate the responses and generate a reference answer for the task.

### Instruction:
{instruction}

### Input:
{input}

### Response 1:
{response 1, generated by a candidate model}

### Response 2:
{response 2, generated by another candidate model}

### Evaluation:
{evaluation result}
{evaluation reason}

### Reference:
{a reference response for the instruction}
```

Figure 6: The prompt for training PandaLM.

## A  TRAINING PROMPT DETAILS

We introduce the detailed prompt of training PandaLM in Figure 6.

## B  DIRECTED ACYCLIC GRAPH DEPICTING THE MIXTURE RANKING OF MODELS TRAINED USING BOTH ALPACA'S AND PANDALM'S HYPERPARAMETERS.

A directed acyclic graph (DAG) is presented in Figure 7, illustrating the relative rankings of various models fine-tuned with different sets of hyperparameters. Notably, this ranking differs from those in Figure4, due to the variance in the test data: the test data for 7 is a sampled subset from that used in Figure4 which is deliberately chosen to ensure a high Inter-Annotator Agreement (IAA). A discernible pattern emerges from the rankings: models fine-tuned using PandaLM's hyperparameters consistently outshine their counterparts fine-tuned with Alpaca's. The top-rated model is PandaLM-LLaMA, followed by Alpaca-LLaMA, PandaLM-Bloom, PandaLM-Pythia, PandaLM-OPT, PandaLM-Cerebras-GPT, Alpaca-OPT, Alpaca-Bloom, Alpaca-Pythia, and Alpaca-Cerebras-GPT, in descending order of performance. This juxtaposition accentuates the effectiveness of PandaLM's hyperparameter selection in improving model performance, as models optimized with PandaLM consistently rank higher than those using Alpaca's hyperparameters in the hybrid ranking. These findings underscore the potential of PandaLM as a powerful tool in enhancing the performance of large language models, further supporting the assertion of its efficacy.

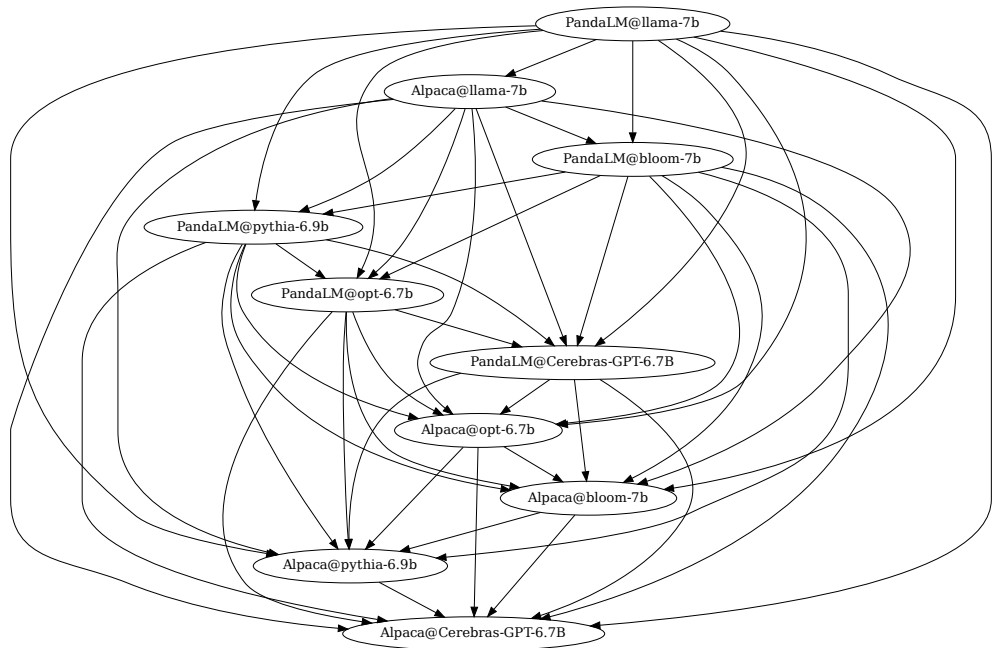

Figure 7: Directed Acyclic Graph depicting the mixture ranking of models trained using both Alpaca's and PandaLM's hyperparameters. The models are ranked from strongest to weakest in the following order: PandaLM-LLaMA, Alpaca-LLaMA, PandaLM-Bloom, PandaLM-Pythia, PandaLM-OPT, PandaLM-Cerebras-GPT, Alpaca-OPT, Alpaca-Bloom, Alpaca-Pythia, Alpaca-Cerebras-GPT.

Table 6: Comparison on several downstream tasks using lm-eval(Gao et al., 2021) between foundation models fine-tuned on Alpaca's hyperparameters, and foundation models fine-tuned with PandaLM. Note that the MMLU task consists of 57 subtasks, which means providing a comprehensive standard deviation here is not feasible.

| | ARC-Challenge-acc_norm(25-shot) | Hellaswag-acc_norm(10-shot) | MMLU-average-acc(5-shot) | TruthfulQA-mc2(0-shot) | Average |
|---|---|---|---|---|---|
| llama-7b original | 0.4923±0.0146 | 0.7583±0.0043 | 0.3306 | 0.3703±0.0141 | 0.4879 |
| llama-7b w/ PandaLM | **0.5162±0.0146** | **0.7764±0.0042** | **0.3396** | **0.3801±0.0145** | **0.5031** |
| opt-6.7b original | **0.3805±0.0142** | 0.6535±0.0047 | 0.2476 | 0.3587±0.0139 | 0.4101 |
| opt-6.7b w/ PandaLM | 0.3771±0.0142 | **0.6540±0.0047** | **0.2502** | **0.3609±0.0142** | **0.4106** |
| pythia-6.9b original | 0.3848±0.0142 | 0.6093±0.0049 | 0.2490 | **0.4187±0.0148** | 0.4155 |
| pythia-6.9b w/ PandaLM | **0.4130±0.0144** | **0.6337±0.0048** | **0.2581** | 0.3972±0.0144 | **0.4255** |
| bloom-7b original | **0.3985±0.0143** | **0.6086±0.0049** | **0.2635** | 0.3975±0.0148 | **0.4170** |
| bloom-7b w/ PandaLM | 0.3951±0.0143 | 0.6084±0.0049 | 0.2520 | **0.3997±0.0149** | 0.4138 |
| Cerebras-GPT-6.7B original | 0.3524±0.0140 | **0.5613±0.0050** | **0.2584** | **0.3624±0.0140** | **0.3836** |
| Cerebras-GPT-6.7B w/ PandaLM | **0.3558±0.0140** | 0.5550±0.0050 | 0.2452 | 0.3448±0.0141 | 0.3752 |

## C  COMPARISONS BETWEEN ORIGINAL MODELS AND MODELS TUNED USING PANDALM ON TRADITIONAL TASKS

We compare fine-tuned LLMs on various traditional tasks with lm-eval (Gao et al., 2021). Although the majority of language models exhibit improved performance after finetuning with PandaLM, Cerebras exhibits a decline. This highlights the importance of nuanced, subjective evaluations (win/tie/lose of responses). Human evaluations, as well as assessments from GPT-4 and GPT-3.5, all concur in indicating a better performance from Cerebras when paired with PandaLM. This is also confirmed in (Yu et al., 2024).

As shown in Table 7, the evaluation results of language models show that lower perplexity, indicating better predictive ability in pretraining or other tasks, does not always mean better overall performance of instruction-tuned models. For example, LLaMA-PandaLM has a higher perplexity than LLaMA-Alpaca but outperforms it in both pairwise comparisons (PandaLM, GPT, Human) and traditional

Table 7: Analysis on perplexity and other evaluation metrics. Note that we report the win rate over 170 samples of PandaLM, GPT, and Human.

| Model | Perplexity (↓) | PandaLM-7B (↑) | PandaLM-70B (↑) | GPT-3.5 (↑) | GPT-4 (↑) | Human (↑) | lm-eval avg. score (↑) |
|---|---|---|---|---|---|---|---|
| LLaMA-Alpaca | **2.75** | 15.88% | 22.94% | 15.29% | 10.00% | 12.35% | 0.4879 |
| LLaMA-PandaLM | 2.81 | **19.41%** | **35.88%** | **26.47%** | **23.53%** | **48.24%** | **0.5031** |

tasks (lm-eval). This suggests that while perplexity is not feasible for instruction-tuned models where lower perplexity might mean overfitting and less generalizability.

## D  LAW / BIOMEDICAL DATASETS INTRODUCTION

Specifically, we assess PandaLM's proficiency using the LSAT (Law School Admission Test) dataset, which serves as an entrance exam question set for American law schools. This dataset incorporates 1,009 questions, further divided into three subsets: AR, LR, and RC. In the realm of biomedicine, we use the PubMedQA dataset—a vast repository for biomedical retrieval QA data, boasting 1k expert annotations, 61.2k unlabeled entries, and a massive 211.3k human-generated QA instances. For our evaluation, we rely on the labeled section (PubMedQA-l) that contains 1k instances. Each instance encompasses a question, context, and label. Additionally, we tap into the BioASQ dataset, specifically leveraging the task b dataset from its 11th challenge. This dataset is renowned for its biomedical semantic indexing and question-answering (QA) capabilities. From it, we use 1k samples for our assessment. We will test code/math dataset Cobbe et al. (2021); Zeng et al. (2022b) in future work.

## E  DATA SIZE AND QUALITY ANALYSIS IN INSTRUCTION TUNING

We conduct an ablation study to investigate the impact of training data size (up to 1,344,000) on the performance of the model, given optimal hyperparameters. Importantly, a relationship exists between the size and quality of training data. Thus, we focus on an ablation study of data size here, but conducting a similar experiment on data quality is feasible. We derive the results from PandaLM-7B. The objective is to discern how much training data is required to reach each model's peak performance. Table 8 reveals the optimal quantity of training data varies among models. More training data typically enhances model performance. However, an optimal point exists for each model, beyond which further data doesn't improve performance. For example, the OPT model peaks at 992,000 data points, indicating additional data does not enhance the model's performance.

Table 8: Optimal training data size for each model.

| Model | Bloom | Cerebras-GPT | LLaMA | OPT | Pythia |
|---|---|---|---|---|---|
| Optimal Training Data Size | 1,216,000 | 1,344,000 | 11,520,000 | 992,000 | 1,344,000 |

## F  LoRA ANALYSIS IN INSTRUCTION TUNING

We further aim to evaluate the efficacy of Low-Rank Adaptation (LoRA) (Hu et al.) compared to full fine-tuning across various models, utilizing optimal hyperparameters. The results are also obtained from PandaLM-7B. Our analysis seeks to provide a comparative understanding of these tuning methodologies. As shown in Table 9, the results for the Bloom model reveal a distinct advantage for full fine-tuning, which triumphs over LoRA in 66 instances as opposed to LoRA's 35. Notably, they tie in 69 instances. In the case of the Cerebras model, full fine-tuning again proves superior, leading in 59 cases compared to LoRA's 40, despite drawing even 71 times. The trend of full fine-tuning superiority is consistent in the LLaMA model. Out of 170 instances, full fine-tuning results in better performance in 48 instances, whereas LoRA emerges victorious in only 28 instances. The majority of the results are tied, amounting to 94 instances. In the OPT model, full fine-tuning once more showcases its advantage with 64 instances of superior performance compared to LoRA's 33, while recording a tie in 73 instances. Lastly, for the Pythia model, full fine-tuning leads the race with 71 instances of better performance against LoRA's 21, and a tie occurring in 78 instances. These results

underscore that full fine-tuning generally yields more favorable results compared to the use of LoRA, though the outcomes can vary depending on the model. Despite the considerable number of ties, full fine-tuning holds the upper hand in most models, thereby highlighting its effectiveness. This suggests that while LoRA may provide comparable results in some instances, a strategy of full fine-tuning often proves to be the more beneficial approach in enhancing model performance.

Table 9: Comparison of LoRA and Full Fine-tuning.

| Model | LoRA Wins | Full Fine-tuning Wins | Ties |
|---|---|---|---|
| Bloom | 35 | 66 | 69 |
| Cerebras-GPT | 40 | 59 | 71 |
| LLaMA | 28 | 48 | 94 |
| OPT | 33 | 64 | 73 |
| Pythia | 21 | 71 | 78 |

## G  LEVERAGING PRE-TRAINED MODELS AND OTHER INSTRUCTION TUNED MODELS FOR EVALUATION

Employing LLMs for response evaluation without additional training is a natural direction for the task. However, implementing evaluation criteria through zero-shot or few-shot methods is challenging for LLMs due to the necessity for extended context lengths.

We have undertaken experiments using zero-shot and few-shot (in-context learning Dong et al. (2022); Yang et al. (2023b)) evaluations with LLaMA. Our observations indicate that an un-tuned LLaMA struggles with adhering to user-specified format requirements. Consequently, our experiments focused on computing and comparing the log-likelihood of generating continuations (e.g., determining whether "Response 1 is better," "Response 2 is better," or if both responses are similar in quality) from the same context. We regard the choice with the highest log-likelihood as the prediction result. We also alternated response order in our experiments to reduce position bias. Furthermore, we undertook experiments with Vicuna, a finetuned version of LLaMA. The experiments demonstrated that the evaluation capabilities of instruction-tuned models possess significant potential for enhancement.

The results in Table 10 highlight the importance of tailored tuning for evaluation, a precisely-tuned smaller model outperforms a larger one in zero and few-shot scenarios.

## H  ENHANCING PANDALM WITH REFINED SUPERVISION.

In our supervision goal, we incorporate not only the comparative result of responses but also a succinct explanation and a reference response. This methodology augments PandaLM's comprehension of the evaluation criteria.

Table 10: Ablation study of directly using pre-trained models and instruction tuned models for evaluation.

| Model | Accuracy | Precision | Recall | F1 score |
|---|---|---|---|---|
| LLaMA-7B 0-shot (log-likelihood) | 12.11 | 70.23 | 34.52 | 8.77 |
| LLaMA-30B 0-shot (log-likelihood) | 31.43 | 56.48 | 43.12 | 32.83 |
| LLaMA-7B 5-shot (log-likelihood) | 24.82 | 46.99 | 39.79 | 25.43 |
| LLaMA-30B 5-shot (log-likelihood) | 42.24 | 61.99 | 51.76 | 42.93 |
| Vicuna-7B (log-likelihood) | 15.92 | 57.53 | 34.90 | 14.90 |
| Vicuna-13B (log-likelihood) | 35.24 | 57.45 | 43.65 | 36.29 |
| PandaLM-7B | **59.26** | 57.28 | 59.23 | 54.56 |
| PandaLM-7B (log-likelihood) | **59.26** | **59.70** | **63.07** | **55.78** |

Table 11: Ablation study of supervision goal.

| Model | Accuracy | Precision | Recall | F1 |
|---|---|---|---|---|
| PandaLM-7B (with only eval label) | 0.4725 | 0.4505 | 0.4725 | 0.3152 |
| PandaLM-7B | 0.5926 | 0.5728 | 0.5923 | 0.5456 |

To empirically gauge the significance of this explanation, an experiment was executed. Here, the explanation and reference were omitted during training, and only the categorical outcomes (0/1/2 or Tie/Win/Lose) were retained in the dataset for training a fresh iteration of PandaLM. The results, as depicted in Table 11, demonstrate that in the absence of the explanation, PandaLM encounters difficulties in precisely determining the preferable response.

## I    HUMAN EVALUATION DATASHEET

We employ human annotators from a crowdsourcing company and pay them fairly. In particular, we pay our annotators 50 dollars per hour, which is above the average local income level. We have filled out the Google Sheet provided in (Shimorina & Belz, 2022).

## J    HYPERPARAMETER OPTIMIZATION ANALYSIS

In our hyperparameter searching process, we explored a range of learning rates, epochs, optimizers, and schedulers. The learning rates tested varied from 2e-6 to 2e-4, with model checkpoints saved at the end of each epoch. Performance was rigorously assessed through pairwise comparisons between checkpoints, counting the win rounds for each model, as detailed in Figure 8.

Our analysis, as depicted in Figure 8a, suggests a tendency towards a learning rate of 2e-5, although this preference was not uniformly clear across all models. Figure 8b demonstrates the variability in the optimal number of epochs, with a trend showing that peak performance often occurs around the fourth or fifth epoch. This evidence points to the complex interplay of hyperparameters with model performance, which is further influenced by data distribution, optimizer, and scheduler choices.

The findings from our hyperparameter optimization process highlight that there is no universally optimal setting for different models and training setups. While a pattern emerged suggesting that a learning rate around 2e-5 and an epoch count near 4 might be beneficial in some cases, these results are not conclusive. This reinforces the need for specific hyperparameter searches for different models, as demonstrated in our visualizations. A tailored approach to hyperparameter optimization is essential, as it allows for a more nuanced understanding of model performance across various scenarios.

Besides, we implemented an early stopping strategy using Pandalm. We focus specifically on LLaMA. Our experiments showed that in some cases, a model's performance at epoch 3 was inferior to that at epoch 2. However, subsequent epochs demonstrated performance improvements. This indicates that early stopping may not always be suitable for large model fine-tuning, as it could prematurely halt training before reaching optimal performance.

Table 12: Analysis of PandaLM's Evaluation Capability on Unseen Models.

| Model Comparison | PandaLM | Human | Metrics (P, R, F1) |
|---|---|---|---|
| llama1-7b vs llama2-7b | (23,61,16) | (23,70,7) | (0.7061, 0.7100, 0.6932) |
| llama1-13b vs llama2-13b | (18,73,9) | (20,68,12) | (0.7032, 0.6800, 0.6899) |
| llama1-65b vs llama2-70b | (20,66,14) | (34,56,10) | (0.7269, 0.6600, 0.6808) |

## K    MODEL SHIFT ANALYSIS

In Table 12, we provide a detailed comparison of PandaLM's performance against human benchmarks and in the context of different versions of instruction-tuned LLaMA models. Note that llama1-13b, llama1-65b and llama2 are indicative of model shift. The results demonstrate that PandaLM aligns

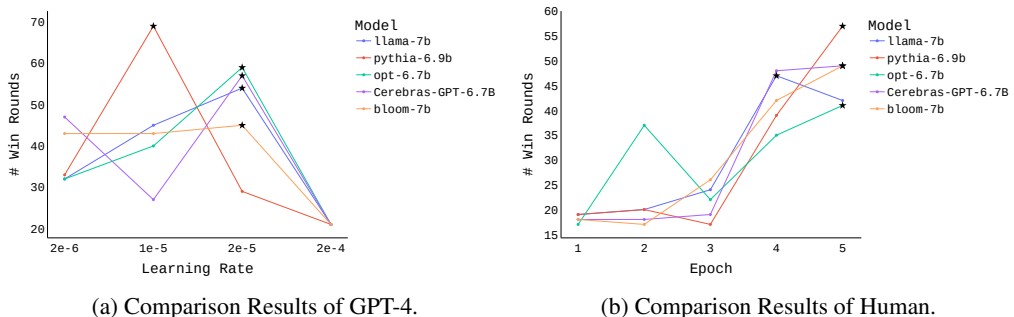

(a) Comparison Results of GPT-4.

(b) Comparison Results of Human.

Figure 8: Hyperparameter Optimization Analysis in PandaLM. The figure illustrates the performance across different learning rates and variability in model performance across epochs.

closely with humans, consistently showing a preference for the LLama-2 model. This alignment is in line with expectations, as LLama-2 benefits from more pre-training data. Such findings highlight the significance of extensive pre-training in developing language models that are more skilled at understanding and correctly responding to various instructions.

