# OpenReview forum: "PandaLM: An Automatic Evaluation Benchmark for LLM Instruction Tuning Optimization"
_ICLR.cc/2024/Conference — ICLR 2024 poster_

### Official Review · Reviewer_8MGB · 2023-11-01

**Soundness:** 2 fair
**Presentation:** 2 fair
**Contribution:** 2 fair
**Rating:** 5
**Confidence:** 4

**Summary:**

- This paper raises a pivotal question: How can we design a local LLM-based evaluator that safeguards privacy and minimizes potential data leakage?
- In this study, evaluations from GPT-3.5 are distilled into a model named PandaLM, which utilizes the capabilities of LLaMA, achieving performance on par with both GPT-3.5 and GPT-4.
- Furthermore, the researchers have curated a robust human-annotated dataset, which is indispensable for verifying PandaLM's effectiveness and paving the way for future studies.
- Additionally, PandaLM has been employed to fine-tune the hyperparameters of various open-source LLMs, leading to significant performance improvements compared to default configurations.

**Strengths:**

- This research innovatively introduces a local LLM-based evaluator, positioning it as a viable alternative to ChatGPT and GPT-4. The emphasis on local models highlights their intrinsic advantage in minimizing the risks associated with data leakage.
- PandaLM, as presented in the paper, exhibits a commendable performance that stands shoulder-to-shoulder with both GPT-3.5 and GPT-4. Notably, the resilience of PandaLM is evident as it continues to demonstrate robustness even when transitioning to the specialized domain of medicine.

**Weaknesses:**

- While the authors utilize PandaLM to optimize hyperparameters, there's an evident absence of evaluation results concerning different hyperparameter settings. Specifically, insights into how performance varies with adjustments in learning rate or epochs would be beneficial.
- The presented evaluation results aren't juxtaposed with learning metrics like loss and perplexity. Despite the authors' assertion that perplexity might offer optimization directions, this comparison is conspicuously missing.
- The hyperparameter settings presented appear incomplete, particularly the omission of early stopping.
- There seems to be a gap in the generalized and robustness evaluation of PandaLM. Although the authors have assessed PandaLM in the context of legal and biological domains (indicative of domain shift), a broader spectrum of evaluations, especially with unseen models like LLaMA-2 (indicative of model shift), would enhance its credibility.
- While evaluating LLMs is undeniably vital, solely focusing on outcome evaluation might be resource-intensive. A deeper dive into behavior prediction could provide a more comprehensive understanding and potentially be more resource-efficient.

**Questions:**

1. How does a model optimized by loss or perplexity fare in evaluation results? Can you shed light on the advantages of PandaLM over automatic metrics in hyperparameter optimization? The statement, "... Perplexity ... does not effectively guide the choice of hyperparameters", prompts questions regarding the mechanism by which PandaLM aids in hyperparameter selection. Is there a direct correlation between PandaLM's results and the choice of hyperparameters? Additionally, could you share the results for models selected based on optimal perplexity?
2. Is there an implementation of the early stopping strategy? If so, how does this impact overall performance?
3. How well does PandaLM generalize in the face of model shifts? There's a latent concern regarding potential overfitting with PandaLM, which might explain its superior performance over GPT-3.5 and GPT-4. It would be valuable to see results on unseen models, i.e., models whose generations weren't part of the training set. For instance, while the dataset includes responses from models like LLaMA-7B, Bloom-7B, and others (deemed as 'seen' models), results on LLaMA-2-7B (an 'unseen' model) are anticipated.
4. Could you clarify the contents and implications of Table-3? How do 'alpaca' and 'vicuna', mentioned in this section, correlate? Moreover, where can we find the results for GPT-3.5-turbo and GPT-4 in the legal and biological domains?
5. Building on the topic of scalable oversight, how does PandaLM perform with larger models, particularly LLaMA-13B and LLaMA-70B? A deeper insight into scalable oversight would be highly appreciated.
6. Given the substantial evaluation costs associated with fine-tuned models, how might one conduct more granular evaluations prior to the fine-tuning process? I'm eager to hear the authors' take on behavior prediction as a potentially promising avenue beyond the scope of PandaLM.

---

> ### Author Response · Authors · 2023-11-16
> **Response to Reviewer 8MGB (Part 1)**
>
> Dear Reviewer,
>
> We sincerely thank you for your insightful comments and valuable suggestions on our manuscript. Your expertise and detailed review have been instrumental in enhancing the quality and clarity of our work. In response to your comments, we have carefully examined and addressed each issue you raised. For your convenience, we have marked all the related revisions in blue within the manuscript, ensuring easy identification of the changes made. We welcome any further inquiries or suggestions you might have, as your input is invaluable in refining our research. Thank you for your continued guidance and support in this process.
>
> ### Weakness 1:
> While the authors utilize PandaLM to optimize hyperparameters, there's an evident absence of evaluation results concerning different hyperparameter settings. Specifically, insights into how performance varies with adjustments in learning rate or epochs would be beneficial.
>
>
>
> ### Answer:
>
>
> Thank you for your insightful comments regarding our utilization of PandaLM for hyperparameter optimization. We have added visualization and analysis in Appendix J of the revised paper. In our hyperparameter searching process, we explored a range of learning rates, epochs, optimizers, and schedulers. The learning rates tested varied from 2e-6 to 2e-4, with model checkpoints saved at the end of each epoch. Performance was rigorously assessed through pairwise comparisons between checkpoints, counting the win rounds for each model. Our analysis, as depicted in the figure, suggests a tendency towards a learning rate of 2e-5, although this preference was not uniformly clear across all models. The figure also demonstrates the variability in the optimal number of epochs, with a trend showing that peak performance often occurs around the fourth or fifth epoch. This evidence points to the complex interplay of hyperparameters with model performance, which is further influenced by data distribution, optimizer, and scheduler choices. The findings from our hyperparameter optimization process highlight that there is no universally optimal setting for different models and training setups. While a pattern emerged suggesting that a learning rate around 2e-5 and an epoch count near 4 might be beneficial in some cases, these results are not conclusive. This reinforces the need for specific hyperparameter searches for different models, as demonstrated in our visualizations. A tailored approach to hyperparameter optimization is essential, as it allows for a more nuanced understanding of model performance across various scenarios.
>
>
>
>
>
> ### Weakness 2:
> The presented evaluation results aren't juxtaposed with learning metrics like loss and perplexity. Despite the authors' assertion that perplexity might offer optimization directions, this comparison is conspicuously missing.
>
>
>
> ### Answer:
>
>
>
> | Model          | Perplexity (&darr;) | PandaLM-7B (&uarr;)  | PandaLM-70B (&uarr;) | GPT-3.5 (&uarr;) | GPT-4 (&uarr;) | Human (&uarr;) | lm-eval average score (&uarr;) |
> |----------------|---------------------|---------------------|----------------------|------------------|----------------|----------------|------------------|
> | LLaMA-Alpaca   | **2.75**                | 15.88%                  |       22.94%       | 15.29%               | 10.00%             | 12.35%             | 0.4879           |
> | LLaMA-PandaLM  | 2.81                | **19.41%**                  |        **35.88%**       | **26.47%**               | **23.53%**             | **48.24%**             | **0.5031**           |
>
>
> Thanks for your valuable suggestions, we have included the analysis of perplexity in Appendix C in the revised paper. We report the win rate of LLaMA-Alpaca and LLaMA-PandaLM over 170 samples. The evaluation results show that lower perplexity, indicating better predictive ability in pretraining, does not always mean better overall performance of instruction tuned models. For example, LLaMA-PandaLM has a higher perplexity than LLaMA-Alpaca but outperforms it in both pairwise comparison (PandaLM,GPT,Human) and traditional tasks (lm-eval). In such contexts, lower perplexity might even suggest potential overfitting, thereby diminishing the model's generalizability.
>
>
>
>
>
> ### Weakness 3:
> The hyperparameter settings presented appear incomplete, particularly the omission of early stopping.
>
>
>
> ### Answer:
>
>
> Thanks for pointing out, we implemented an early stopping strategy using Pandalm. We focus specifically on LLaMA. Our experiments showed that in some cases, a model's performance at epoch 3 was inferior to that at epoch 2. However, subsequent epochs demonstrated performance improvements. This indicates that early stopping may not always be suitable for large model fine-tuning, as it could prematurely halt training before reaching optimal performance. We have added this analysis in Appendix J of the revised paper.

---

> ### Author Response · Authors · 2023-11-16
> **Response to Reviewer 8MGB (Part 2)**
>
> ### Weakness 4:
> There seems to be a gap in the generalized and robustness evaluation of PandaLM. Although the authors have assessed PandaLM in the context of legal and biological domains (indicative of domain shift), a broader spectrum of evaluations, especially with unseen models like LLaMA-2 (indicative of model shift), would enhance its credibility.
>
>
>
> ### Answer:
> | Model Comparison          | PandaLM       | Human         | Metrics (P, R, F1)       |
> |---------------------------|---------------|---------------|--------------------------|
> | llama1-7b vs llama2-7b    | (23,61,16)    | (23,70,7)     | (0.7061, 0.7100, 0.6932) |
> | llama1-13b vs llama2-13b  | (18,73,9)     | (20,68,12)    | (0.7032, 0.6800, 0.6899) |
> | llama1-65b vs llama2-70b  | (20,66,14)    | (34,56,10)    | (0.7269, 0.6600, 0.6808) |
>
> We provide a detailed comparison of PandaLM's performance against human benchmarks and in the context of different versions of instruction tuned LLaMA models. Note that llama1-13b, llama1-65b and llama2 is indicative of model shift. The results clearly demonstrate that PandaLM aligns closely with human, consistently showing a preference for the LLama-2 model. This alignment is in line with expectations, as LLama-2 benefits from more pre-training data. We have added this analysis in Appendix K in our paper.
>
>
> ### Weakness 5:
> While evaluating LLMs is undeniably vital, solely focusing on outcome evaluation might be resource-intensive. A deeper dive into behavior prediction could provide a more comprehensive understanding and potentially be more resource-efficient.
>
>
> ### Answer:
> We agree with your point about the importance of evaluating large language models (LLMs) beyond just outcome-based measures. Currently, our emphasis is primarily on outcome-based evaluation, which is indeed resource-intensive. However, integrating behavior prediction into evaluation framework could offer a more comprehensive understanding of LLM performance. For instance, analyzing and evaluating the extended text outputs of an untuned LLM can help predict how a tuned version might behave in various scenarios. This approach could provide a more efficient and insightful way to balance resource-heavy outcome assessments. We have added this in the Limitation Section of the revised paper.
>
> ### Question 1:
> How does a model optimized by loss or perplexity fare in evaluation results? Can you shed light on the advantages of PandaLM over automatic metrics in hyperparameter optimization? The statement, "... Perplexity ... does not effectively guide the choice of hyperparameters", prompts questions regarding the mechanism by which PandaLM aids in hyperparameter selection. Is there a direct correlation between PandaLM's results and the choice of hyperparameters? Additionally, could you share the results for models selected based on optimal perplexity?
>
> ### Answer:
> The experimental results in response to Weakness 2 indicate that model optimized by loss or perplexity might not always correlate with high performance in practical tasks. While perplexity measures predictive ability during pretraining, it doesn't encompass all aspects of language understanding crucial in practical applications. Therefore, PandaLM's broader assessment is more effective for hyperparameter optimization, as it evaluates models in complex, instruction-based scenarios. We have included the analysis of perplexity in Appendix C in the revised paper.
>
> ### Question 2:
> Is there an implementation of the early stopping strategy? If so, how does this impact overall performance?
> ### Answer:
> We implemented an early stopping strategy using Pandalm. Due to time constraints, we focus specifically on LLaMA. We use Pandalm to determine if the current training checkpoint surpassed the previous one. If the current checkpoint was found to be inferior, the training process was stopped. It turned out that LLaMA training stopped at epoch 5, aligning with our earlier observations from a comprehensive pairwise comparison of all tuned LLaMA checkpoints, which also considered variations in learning rate, optimizer, and scheduler. The early stopping strategy proved effective in reducing the time required to identify the best epochs for the model. However, it's important to note that this approach still necessitates experimentation with different learning rates, optimizers, and schedulers to optimize performance fully. We have added the analysis of early stopping in Appendix J of the revised paper.

---

> > ### Author Response · Authors · 2023-11-16
> > **Response to Reviewer 8MGB (Part 3)**
> >
> > ### Question 3:
> > How well does PandaLM generalize in the face of model shifts? There's a latent concern regarding potential overfitting with PandaLM, which might explain its superior performance over GPT-3.5 and GPT-4. It would be valuable to see results on unseen models, i.e., models whose generations weren't part of the training set. For instance, while the dataset includes responses from models like LLaMA-7B, Bloom-7B, and others (deemed as 'seen' models), results on LLaMA-2-7B (an 'unseen' model) are anticipated.
> >
> > ### Answer:
> > The experimental results in response to Weakness 4 show that PandaLM has notable generalization capabilities in the context of model shifts. The provided comparisons between various unseen models reveal that PandaLM aligns closely with human benchmarks. This is particularly important as it suggests PandaLM's effectiveness is not limited to models it was trained with ('seen' models). Instead, it maintains a high level of performance even when evaluating generations from models not included in its training set. This ability to generalize across different models reinforces PandaLM's utility as a versatile and reliable evaluation tool for LLMs. We have added the analysis in Appendix K in our paper.
> >
> >
> > ### Question 4:
> > Could you clarify the contents and implications of Table-3? How do 'alpaca' and 'vicuna', mentioned in this section, correlate? Moreover, where can we find the results for GPT-3.5-turbo and GPT-4 in the legal and biological domains?
> >
> > ### Answer:
> > In Table 3 of our study, we address the challenge of evaluating responses in the legal and biological domains. Due to the difficulty in hiring bio and legal experts for annotation, instead of relying on human annotators, GPT-4 is utilized to produce the gold standard answers for the responses. The comparison involves analyzing the performance of 'vicuna' and 'alpaca' on datasets such as LSAT, PubMedQA, and BioASQ. In this context, PandaLM acts as the evaluator, and GPT-4's evaluations are used as the standard for comparison. Given that we are dealing with a three-class classification problem (win, tie, lose), a random guess would typically result in approximately 33% accuracy in terms of precision, recall, and F1 score. The results much higher than 33% show that PandaLM demonstrates strong consistency with GPT-4's evaluations. This is further validated through a human evaluation conducted on a subset of 30 samples from the BioASQ dataset. The human evaluation revealed that both PandaLM and GPT-4 tended to favor Vicuna over Alpaca, indicating a consistent trend in their evaluations. We have modified the analysis in Section 4 in our revised paper.
> >
> > ### Question 5:
> > Building on the topic of scalable oversight, how does PandaLM perform with larger models, particularly LLaMA-13B and LLaMA-70B? A deeper insight into scalable oversight would be highly appreciated.
> >
> > ### Answer:
> > The results presented in response to Weakness 4 indicates that PandaLM performs effectively when evaluating larger language models, such as LLaMA-13B and LLaMA-70B. The evaluation of PandaLM indeed demonstrates that as the pre-training data for instruction-tuned models increases, their ability to understand and follow instructions improves. Such findings highlight the significance of extensive pre-training in developing language models that are more skilled at understanding and correctly responding to various instructions. We have added the analysis in Appendix K in our paper.
> >
> > ### Question 6:
> > Given the substantial evaluation costs associated with fine-tuned models, how might one conduct more granular evaluations prior to the fine-tuning process? I'm eager to hear the authors' take on behavior prediction as a potentially promising avenue beyond the scope of PandaLM.
> >
> > ### Answer:
> > We agree with your point about the importance of conducting more granular evaluations prior to the fine-tuning process. Analyzing and evaluating the extended text outputs of an untuned LLM can help predict how a tuned version might behave in various scenarios. This approach could provide a more efficient and insightful way to balance resource-heavy outcome assessments with. We have added this in the Limitation Section of the revised paper.

---

### Official Review · Reviewer_FMz2 · 2023-11-01

**Soundness:** 3 good
**Presentation:** 3 good
**Contribution:** 3 good
**Rating:** 8
**Confidence:** 3

**Summary:**

The authors introduce PandaLM, a language model that can be used to evaluate and compare the response of other language models. The evaluation criteria are supposed to centre around important but subjective factors such as conciseness, adherence to instructions, and comprehensiveness. This is important because in contrast to tasks like classification, the responses of a language model cannot be justly evaluated using a static set. The alternative methods, e.g. human evaluation or using expensive cloud LLMs through APIs, are also less than ideal.

The authors use this trained model to not only evaluate other instruction tuned language models from the Alpaca family, but also use it to find better hyperparameters for training them. To validate their hypothesis, they show that 1) the evaluations from PandaLM match human evaluations to a greater degree than GPT-3.5 and GPT-4; and 2) they also show that the resulting hyper-parameter-optimized models' performance are superior to their predecessors as evaluated by human evaluators, GPT-3.5, and GPT-4.

I believe they also plan to make the dataset used to evaluate PandaLM itself available to the open source community as well.

**Strengths:**

- The problem being addressed is certainly an important one, and the introduction of a repeatable process with reduced cost (as opposed to human evaluation and cloud LLMs) will definitely help the community and the research of LLMs
- Open sourcing the model and the dataset (I assume)
- The 70B model beating GPT-4, definitely makes it a worthwhile method
- The majority of the results seem to point in the same direction, which I find to be rare and makes it easier to draw conclusions from the work

- Originality: I do not know of a similar attempt to train such a model, and hence assume the work to be original, even though there isn't a lack of people who have tried to use LMs to evaluate other LMs
- Clarity: the description seemed easy to understand and without any unnecessary math / jargon to complicate things
- Significance: though I expect the proposed models to start becoming obsolete rather soon due to the speed of innovation, I still believe the model could be a strongly positive force in solving the problem of LLM evaluation

**Weaknesses:**

- Even though this model is still better than GPT-4 which is great, still the low accuracy/F1 scores leave one desiring a better model. Can the authors propose a higher-bound on the expected F1/accuracy scores? For example, can we compare this to the percentage of human evaluators usually that agree with each other on the same task?

- In Table 3, PandaLM-7B is shown to have dangerously low numbers: 47% for LSAT and 51% for BioASQ. Assuming that chance accuracy is 50%, this highly questions the validity of this version of the model. I understand that the judge for these results has been GPT-4, however, that doesn't resolve the issue. I highly suggest to not simply dismiss the results, and further validate why. If it turns out that human evaluators actually do prefer this model, then all the better. Otherwise, the 7B version of the model might not be fit to the task of evaluating results outside the domain in which it was trained on.

**Questions:**

I have already asked two questions in the previous section, but I'll include a few minor areas of improvement here.

Minor issues:
- In Table 2, I believe GPT-4's Recall should be highlighted as it's higher than PandaLM-70B's (68% vs 66%)
- In the conclusion, the authors mention that "based on PandaLM evaluations", the superior models are the ones that have been trained using PandaLM. Then they claim that "this order emphasizes the efficacy of PandaLM ...". I believe this claim is incorrect since by definition a model that the PandaLM preferred in hyperparameter tuning, is already preferred to the base model. Hence, one cannot make any claim based on the results. If I misunderstood, then please clarify. Otherwise I believe this sentence should be ommitted.

---

> ### Author Response · Authors · 2023-11-16
> **Response to Reviewer FMz2**
>
> Dear Reviewer,
>
> We sincerely thank you for your insightful comments and valuable suggestions on our manuscript. Your expertise and detailed review have been instrumental in enhancing the quality and clarity of our work. In response to your comments, we have carefully examined and addressed each issue you raised. For your convenience, we have marked all the related revisions in blue within the manuscript, ensuring easy identification of the changes made. We welcome any further inquiries or suggestions you might have, as your input is invaluable in refining our research. Thank you for your continued guidance and support in this process.
>
> ### Weakness 1:
> Even though this model is still better than GPT-4 which is great, still the low accuracy/F1 scores leave one desiring a better model. Can the authors propose a higher-bound on the expected F1/accuracy scores? For example, can we compare this to the percentage of human evaluators usually that agree with each other on the same task?
>
>
>
> ### Answer:
> Our test set was annotated by three individuals with inter-annotator agreement (IAA) exceeding 0.85. To refine the model's performance assessment compared to human evaluators, we can use the inter-annotator agreement (IAA) of 0.85 as a benchmark. If our model exceeds this, it indicates strong performance. However, setting a realistic target slightly above this human IAA, say around 0.90, offers a challenging yet achievable goal. We have added this analysis in Section 4 in our revised paper.
>
>
> ### Weakness 2:
> In Table 3, PandaLM-7B is shown to have dangerously low numbers: 47% for LSAT and 51% for BioASQ. Assuming that chance accuracy is 50%, this highly questions the validity of this version of the model. I understand that the judge for these results has been GPT-4, however, that doesn't resolve the issue. I highly suggest to not simply dismiss the results, and further validate why. If it turns out that human evaluators actually do prefer this model, then all the better. Otherwise, the 7B version of the model might not be fit to the task of evaluating results outside the domain in which it was trained on.
>
> ### Answer:
>
> We're addressing a three-category classification task (win/lose/tie), where a random guess would lead to around 33% in precision, recall, and F1 score. PandaLM-7B's results of 47% and 51% are notably above this level. To further validate PandaLM-7B, we conducted a human evaluation with 30 samples on the BioASQ evaluation. The human evaluation showed that both PandaLM-7B and GPT-4 tended to favor Vicuna over Alpaca, indicating a consistent trend in their evaluations. We have added this analysis in Section 4 in our revised paper.
>
>
> ### Question 1:
> In Table 2, I believe GPT-4's Recall should be highlighted as it's higher than PandaLM-70B's (68% vs 66%)
>
> ### Answer:
>
> Thank you for highlighting this detail. We have updated our paper accordingly, marking the revision in blue font for clarity and emphasis.
>
> ### Question 2:
> In the conclusion, the authors mention that "based on PandaLM evaluations", the superior models are the ones that have been trained using PandaLM. Then they claim that "this order emphasizes the efficacy of PandaLM ...". I believe this claim is incorrect since by definition a model that the PandaLM preferred in hyperparameter tuning, is already preferred to the base model. Hence, one cannot make any claim based on the results. If I misunderstood, then please clarify. Otherwise I believe this sentence should be ommitted.
>
> ### Answer:
> We acknowledge the potential oversight in our initial interpretation of the evaluation results. To address this and provide a more accurate representation, we have revised the section in the paper as follows:
>
> "Besides, as shown in Appendix B, based on PandaLM's evaluation, the model demonstrating superior performance is LLaMA-PandaLM. Note that the base foundation model's characteristics can be a significant factor in performance, as evidenced by LLaMA models securing the top two positions. The ranking pattern observed aligns closely with the base model rankings presented in Figure 4."

---

> > ### Comment · Reviewer_FMz2 · 2023-11-23
> > **Acknowledgement of response**
> >
> > I thank the authors for their responses and the modifications. I have no further questions to ask.

---

### Official Review · Reviewer_Vuch · 2023-11-05

**Soundness:** 3 good
**Presentation:** 3 good
**Contribution:** 3 good
**Rating:** 8
**Confidence:** 3

**Summary:**

The paper presents a novel framework for the evaluation of instruction-tuned large language models (LLMs) named PandaLM, which can be used to select the superior one given several LLMs. Rather than focusing on the traditional metrics, like F1, precision or recalls, PandaLM addresses important subjective factors, such as relative conciseness, clarity, adherence to instructions, comprehensiveness, and formality.  Further, the authors also collect a human-annotated test dataset to evaluate the reliability of PandaLM. Finally, the authors use PandaLM to tune several state-of-arts LLMs and yield better performance.

**Strengths:**

1, Build an open-source LLM, PandaLM, which can help to do evaluation of the performance of LLMs on several subjective factors and the performance of PandaLM on a human-annotated dataset. In detail, PandaLM stands out by assessing performance based on a range of subjective factors such as conciseness, clarity, instruction adherence, comprehensiveness, and formality. These aspects are often neglected in traditional benchmarks that focus predominantly on objective correctness.

2, The paper performs extensive experiments to show the effectiveness of PandaLM.  The paper conducts thorough experiments to validate the effectiveness of PandaLM. By benchmarking against industry leaders like GPT-3.5 and GPT-4, the authors provide compelling evidence of PandaLM's capability to serve as a reliable evaluation tool.


3, The paper is well-organized and well-written.

**Weaknesses:**

1, In the abstract and introduction sections,  the authors should make clear the claim point and avoid overclaiming the contricution. For example, in the abstract, rather than simply claiming that PandaLM-7B offers a performance comparable to both GPT-3.5 and GPT-4. Impressively, PandaLM-70B surpasses their performance, the authors should point out it is on the evaluation of the dataset collected by the authors. In the introduction section, instead of claiming that Tuning models with PandaLM-selected hyperparameters yields more substantial performance enhancements, it should point out specifically the performance compared to the LLM searched by Alpaca

2, In Table 1, it is better to add the performance of LLaMA as a judge. By doing so, it can rule out the possibility of the performance of PandLM is caused by the backbone model, rather than the proposed LLM tuning framework.

**Questions:**

see the questions in  Weakness section

---

> ### Author Response · Authors · 2023-11-16
> **Response to Reviewer Vuch**
>
> Dear Reviewer,
>
> We sincerely thank you for your insightful comments and valuable suggestions on our manuscript. Your expertise and detailed review have been instrumental in enhancing the quality and clarity of our work. In response to your comments, we have carefully examined and addressed each issue you raised. For your convenience, we have marked all the related revisions in blue within the manuscript, ensuring easy identification of the changes made. We welcome any further inquiries or suggestions you might have, as your input is invaluable in refining our research. Thank you for your continued guidance and support in this process.
>
> ### Weakness 1:
> In the abstract and introduction sections, the authors should make clear the claim point and avoid overclaiming the contricution. For example, in the abstract, rather than simply claiming that PandaLM-7B offers a performance comparable to both GPT-3.5 and GPT-4. Impressively, PandaLM-70B surpasses their performance, the authors should point out it is on the evaluation of the dataset collected by the authors. In the introduction section, instead of claiming that Tuning models with PandaLM-selected hyperparameters yields more substantial performance enhancements, it should point out specifically the performance compared to the LLM searched by Alpaca.
>
> ### Answer:
> Thank you for your valuable feedback. We have revised the abstract and introduction sections to provide a clearer context for our contributions.
>
> ### Weakness 2:
> In Table 1, it is better to add the performance of LLaMA as a judge. By doing so, it can rule out the possibility of the performance of PandaLM is caused by the backbone model, rather than the proposed LLM tuning framework.
>
> ### Answer:
>
> | **Model**                         | **Accuracy** | **Precision** | **Recall** | **F1 score** |
> |-----------------------------------|--------------|---------------|------------|--------------|
> | LLaMA-7B 0-shot (log-likelihood)  | 12.11        | 70.23         | 34.52      | 8.77         |
> | LLaMA-30B 0-shot (log-likelihood) | 31.43        | 56.48         | 43.12      | 32.83        |
> | LLaMA-7B 5-shot (log-likelihood)  | 24.82        | 46.99         | 39.79      | 25.43        |
> | LLaMA-30B 5-shot (log-likelihood) | 42.24        | 61.99         | 51.76      | 42.93        |
> | Vicuna-7B (log-likelihood)        | 15.92        | 57.53         | 34.90      | 14.90        |
> | Vicuna-13B (log-likelihood)       | 35.24        | 57.45         | 43.65      | 36.29        |
> | PandaLM-7B                        | **59.26**        | 57.28         | 59.23      | 54.56        |
> | PandaLM-7B (log-likelihood)       | **59.26**        | **59.70**         | **63.07**      | **55.78**        |
>
>
> We have acknowledged the significance of incorporating LLaMA's performance as a benchmark. In fact, Appendix G of our manuscript already provides evidence indicating that tailored tuning significantly affects performance in both zero and few-shot scenarios. Our results reveal that a smaller model, when well-tuned, can surpass a larger model that hasn't been fine-tuned. To improve clarity, we have updated Section 5 of our manuscript to provide a detailed explanation of the findings in Appendix G. Apart from utilizing pretrained LLaMA for evaluation, we also conducted experiments with finetuned LLaMA (Vicuna). These experiments showed that the evaluation capabilities of instruction-tuned models can be substantially improved. It's important to note that since Vicuna doesn't always produce evaluation responses in accordance with instructions, we compared the log-likelihood of next token(win/tie/lose) to conduct our experiments.

---

### Meta-Review · Area_Chair_rV4w · 2023-12-05

**Metareview:**

The authors propose pandaLM, an automatic evaluator for instruction-tuned LLMs.  To build this, they collect a human-annotated dataset of human preferences and fine-tune a LM on this dataset. They show PandaLM matches GPT4 despite the much smaller size, and enables easier automated evaluations.

Strengths: this is an important problem for a research community, and the model would help researchers study instruction tuning.

Weakness: the OOD eval is less extensive (though the authors added to this in the rebuttal)

**Justification For Why Not Higher Score:**

This one is probably a borderline spotlight. The results are quite strong, but they're also mostly in-domain results (as pointed out by one reviewer) and the authors added an OOD result, but this probably should be more extensive to make this work really convincing as a general alternative to GPT4. There have been many past reference-free evaluation methods that had very high correlation in-domain, only to show issues under distribution or model shift.

**Justification For Why Not Lower Score:**

The paper is clearly a reasonably strong paper. It should definitely not be rejected.

---

### Decision · Program_Chairs · 2024-01-16

Accept (poster)